# Genetic linkage mapping and quantitative trait locus (QTL) analysis of sweet basil (*Ocimum basilicum* L.) to identify genomic regions associated with cold tolerance and major volatiles

**Lara J. Brindisi**[ID], **Robert Mattera, III**[ID], **Sonika Mudiyala**[ID], **Joshua Honig, James E. Simon**[ID]*

New Use Agriculture and Natural Plant Products Program, Department of Plant Biology, Rutgers University, New Jersey, United States of America

* jimsimon@rutgers.edu

**Data Availability Statement:** The raw reads underlying the results presented in the study are available at the National Center for Biotechnology

## Abstract

Chilling sensitivity is one of the greatest challenges affecting the marketability and profitability of sweet basil (*Ocimum basilicum* L.) in the US and worldwide. Currently, there are no sweet basils commercially available with significant chilling tolerance and traditional aroma profiles. This study was conducted to identify quantitative trait loci (QTLs) responsible for chilling tolerance and aroma compounds in a biparental mapping population, including the Rutgers advanced breeding line that served as a chilling tolerant parent, 'CB15', the chilling sensitive parent, 'Rutgers Obsession DMR' and 200 $F_2$ individuals. Chilling tolerance was assessed by percent necrosis using machine learning and aroma profiling was evaluated using gas chromatography (GC) mass spectrometry (MS). Single nucleotide polymorphism (SNP) markers were generated from genomic sequences derived from double digestion restriction-site associated DNA sequencing (ddRADseq) and converted to genotype data using a reference genome alignment. A genetic linkage map was constructed and five statistically significant QTLs were identified in response to chilling temperatures with possible interactions between QTLs. The QTL on LG24 (*qCH24*) demonstrated the largest effect for chilling response and was significant in all three replicates. No QTLs were identified for linalool, as the population did not segregate sufficiently to detect this trait. Two significant QTLs were identified for estragole (also known as methyl chavicol) with only *qEST1* on LG1 being significant in the multiple-QTL model (MQM). *QEUC26* was identified as a significant QTL for eucalyptol (also known as 1,8-cineole) on LG26. These QTLs may represent key mechanisms for chilling tolerance and aroma in basil, providing critical knowledge for future investigation of these phenotypic traits and molecular breeding.

Information (NCBI) Sequence Read Archive (SRA) under the BioProject accession number: PRJNA1025096 (https://www.ncbi.nlm.nih.gov/sra/PRJNA1025096). All other relevant data are within the paper and its Supporting Information files.

**Funding:** This research was supported by Research Grant No. IS-5292-20 RF from BARD, the United States - Israel Binational Agricultural Research and Development Fund (www.bard-isus.com/), and the New Jersey Agricultural Experiment Station and HATCH project 12170 (www.nifa.usda.gov/grants/programs/hatch-act-1887-multistate-research-fund) awarded to James E. Simon (JES).

**Competing interests:** The authors have declared that no competing interests exist.

# 1. Introduction

Basil (*Ocimum* spp.) is the most commercially important culinary herb in the U.S. and in cuisines around the world. Sweet basil is the most common type of basil grown in the US and Europe and can be marketed as a fresh, frozen or dried product [1]. It is categorized as an ultra-niche, high-value annual crop with the global basil market valued at $57 million in 2020 and expecting to reach $63 million by the end of 2027 [2]. The vast majority of sweet basil produced is marketed as fresh-cut where freshness, texture, shelf life, aroma and taste are crucial.

Sweet basil (*O. basilicum* L.) is recognized as an allotetraploid [3–10]. The total number of chromosomes has been observed to vary, suggesting a complex history of polyploidy and aneuploidy [3, 5, 6, 9, 11–13]. The DNA content is estimated to be 3.9–4.7 pg/2C [9, 14] with a genome size of 2.13 Gb [10]. The genome is characterized as highly repetitive, mainly due to long-terminal repeats [10]. Ample work has been done to understand the genetics of basil, including cytological screening [4, 7–9, 12, 13], karyotyping [3, 5, 13], flow cytometry [9, 14, 15], RAPD, AFLP and SSR analyses [9, 16], de novo linkage mapping [16], RNA-sequencing [17, 18], association mapping [19] and construction of a draft reference genome [10]. Furthermore, our team recently assembled a draft chromosome-level reference genome for *O. basilicum* and will soon be releasing the corresponding annotation [20]. Despite these advancements, much remains to be understood about the genetics of basil, including but not limited to, the potential existence of sub-genomes and their interactions, modes of genetic inheritance and mechanisms for polyploidy and aneuploidy events.

The *Ocimum* genus, and the Lamiaceae or "mint" family in general, are clades with a wide diversity of highly aromatic plants. *O. basilicum* includes plants that not only produce the traditional sweet basil aroma, but also licorice, cinnamon, clove, lemon and lime aromas and more [21–24]. Descriptive analyses with trained panels have identified volatile compounds responsible for the key aromas in *O. basilicum* by correlating sensory evaluations with chemistry analysis [21, 22]. These compounds can be objectively detected and quantified via gas chromatography (GC) mass spectrometry (MS) by liquid injection with the essential oils [22, 23, 25, 26], or by headspace (HS) with dried [21, 26] or fresh plant material [24]. HS is quite useful for relative quantification due to its operational simplicity and ability to avoid hydrodistillation, which can alter the composition of essential oils [26, 27]. HS can be coupled with solid phase microextraction (SPME) to increase detection sensitivity and the number of compounds detected [26, 28] with the caveat that the SPME fiber is not uniformly sensitive to all compounds [29]. HS-SPME has become a common method for volatile analysis due to its ability to detect minor compounds and has been successfully used to identify aroma compounds in basil [30–32].

Major aromatic compounds produced by sweet basil include estragole (also known as methyl chavicol and *p*-allylanisole), eucalyptol (also known as 1,8-cineole), linalool and eugenol. Linalool and eucalyptol belong to the monoterpene and monoterpenoid classes, respectively. Their biosynthetic pathways differ in the composition of their monoterpene synthases, but they are both derived from the same starting compound, geranyl diphosphate produced from the mevalonate pathway [33, 34]. However, estragole is a phenylpropene derived from phenylalanine produced from the phenylpropanoid pathway, while eugenol and methyl eugenol are phenylpropanoids derived from guaiacol [35]. Basil plants naturally produce aroma compounds from both biochemical pathways, concurrently, which is a unique trait for many aromatic plant genera in the Lamiaceae family.

Basil is native to hot, humid environments with its primary center of diversity being in tropical Africa, and secondary centers in tropical Asia and tropical/subtropical South America [1, 36, 37]. Similar to many other plants with tropical origins, basil is "chilling sensitive" with the risk of temperatures less than 12°C (54°F) causing injury and temperatures less than 10°C

(50˚F) inducing severe if not total injury of basil leaves [38–40]. Chilling injury symptoms include leaf necrosis, which appears as leaf spotting or browning, wilting or loss of leaf turgidity, and decay [40–43]. Chilling sensitivity in basil is a major problem for distributers as it is typically shipped at low temperatures with other herbs to minimize disease and decay and costly separate shipping arrangements [41]. Other herbs tend to tolerate these low temperatures, but basil does not, which leads to large post-harvest losses [44].

No Genovese-style sweet basils (*O. basilicum*) on the commercial market have been found to be significantly chilling tolerant to date. Genovese represents the "gold standard" of sweet or Italian basil. The aroma is complex yet distinctly floral, spicy and clove-like. The flowers are white and the leaves are large, glossy and green with a convex cross section. The basil varieties that claim to be cold tolerant have camphor, licorice or pine aromas, with morphological characteristics referred to as "ornamental", such as purple flowers and smaller, matte, purple-green and flat leaves. There are management strategies that help reduce chilling injury in basil [44], including harvesting in the afternoon [40, 41], acclimating plants with less severe low temperatures [39, 40], supplementing with artificial lighting [45, 46] and packaging samples in low density polyethylene bags [39, 47]. However, these approaches have limited success and growers, processors and distributors still suffer sizable economic losses with these added expenses.

Our basil breeding team sought to develop a chilling tolerant basil to fill the market need [42]. Over 6,000 seeds of 'Italian Large Leaf' basil were screened for chilling tolerance. Chilling tolerance was hypothesized to be associated with estragole as most of the chilling basil (CB) lines released a strong licorice aroma. Among the advanced Rutgers breeding lines, 'CB15' was later identified as having the most chilling tolerance of the CB lines under both laboratory and simulated commercial practices, however, further breeding efforts were needed to obtain a chilling tolerant basil with Genovese-style aroma [43]. Thus, this basil was crossed with 'RU Obsession DMR' in the present study to not only impart its Genovese-style aroma on the progeny, but also its downy mildew resistance (DMR), *Fusarium oxysporum* f. sp. *basilica* (FOB) resistance, moderate leaf size and late flowering time.

In the present study, we employed double digestion restriction-site associated DNA sequencing (ddRADseq) to construct a genetic linkage map for an $F_2$ mapping population from a controlled cross between 'RU Obsession DMR' and 'CB15'. The goals of this research were to identify quantitative trait loci (QTLs) related to chilling tolerance and key volatile compounds and to determine if chilling response was associated with aroma. The $F_2$ progeny segregated for chilling tolerance and two aroma compound concentrations, enabling QTL detection for chilling tolerance, estragole and eucalyptol. The genetic linkage map reported herein represents a valuable resource for future genetic exploration of chilling tolerance and aroma in *O. basilicum*.

## 2. Materials and methods

### 2.1 Plant material

Two inbred basil lines (*Ocimum basilicum* L.) were selected as parents: 1) the commercial variety 'Rutgers Obsession DMR' (♀) and 2) a breeding line from Rutgers University developed for cold tolerance known as 'CB15' (♂). 'Rutgers Obsession DMR' was selected for its desirable sweet basil aroma profile, DMR, FOB resistance, moderate leaf size and late flowering time, but is sensitive to cold injury akin to other commercial basil varieties. 'CB15' was selected for its cold tolerance, but is rich in estragole and lacks a traditional sweet basil aroma, has large leaves and no DMR or FOB resistance and flowers early. The parents were crossed to obtain an individual $F_1$ hybrid, which was then selfed to obtain 200 $F_2$ progeny as the Obsession x Chilling Basil (OCB) mapping population. $F_2$ seeds were sown in 128-cell trays with standard

potting soil on May 3, 2021 and transplanted into 3.5" square pots after 6 weeks. Plants were transplanted into round 7.6 L (2 gal) pots 9 weeks after sowing and pruned to maintain a vegetative growth stage. The parents, $F_1$ hybrid and $F_2$ progeny were maintained as mature plants in the Rutgers New Jersey Agricultural Experiment Station (NJAES) greenhouses in New Brunswick, NJ. Plants were grown in natural light supplemented with 400W, 208V high pressure sodium (HPS) HID lamps (P.L. Light Systems, Beamsville, ON, Canada) on a 18 h/6 h light/dark photoperiod with ~27°C/20°C day/night cycle at 70% relative humidity. Plants were watered daily and fertilized with 20-20-20 biweekly from April to September and 20-10-20 from October to March. An IPM program was implemented with beneficial insects and occasional pesticide applications for pest control.

## 2.2 Chilling assay

Leaves were assessed for necrosis based on the postharvest chilling methods and scoring as described by Brindisi et al. [43]. Healthy leaves were harvested from each mature plant one node below the first set of fully expanded leaves and sealed in a perforated plastic bag. Bags (n = 3) containing 3–5 leaves were collected for each parent, $F_2$ plant and the $F_1$ hybrid. One bag from each plant was randomly placed in a cardboard box in a randomized complete block design (RCBD). The three cardboard boxes were treated as blocks and randomly arranged in a walk-in refrigerator (Mr. Winter, 1992, 593.7 ft$^3$) in the NJAES greenhouse. The leaves were refrigerated for 4 days at 3–5°C. The entire CRBD was repeated on three dates: October 7, 2021, December 2, 2021 and February 2, 2022. Basil leaves in each bag were photographed together in photo studio light boxes (Glendan, 12"x12") against a black background. Images were acquired using mobile devices (iPhone 7–12). The percent leaf necrosis was evaluated by the machine learning program, Leaf Necrosis Classifier (LNC) [48, 49] by eliminating the background and comparing the number of pixels of necrotic or brown leaf tissue to the number of pixels of the entire leaf blade. The final percent leaf necrosis value for each of the plants was determined by averaging the blocks in each replicate. A low percent necrosis (0%) represents chilling tolerance while a high percent necrosis (100%) represents chilling sensitivity.

## 2.3 Aroma volatile analysis

Dried and ground leaf samples from the $F_2$ plants, parents and $F_1$ were analyzed in triplicate (n = 3) using a Shimadzu Gas Chromatograph (GC) 2010 Plus Series via Headspace Solid Phase Microextraction (SPME) with the Shimadzu TQ8040 Triple-Q Mass Spectrometer (MS) and AOC 6000 autosampler (Shimadzu, Kyoto, Japan). Injections were extracted for 5 min at 250°C and desorbed for 2 min with an analysis time of 19 min. Samples were subject to a 5 split with a sampling time of 1 min. Helium was used as the carrier gas at a pressure of 47.7 kPA with a total flow of 2 mL/min and a column flow of 1 mL/min. The column was an SH-Rxi-5Sil MS fused silica capillary column of 0.25 um thickness, 30 m length and 0.25 mm diameter. The column oven temperature was set to 35°C for 4 min and then to 250°C at a rate of 20°C/min and held for 2 min. The ion source temperature was 200°C and the interface temperature was 250°C. Mass spectra were observed from 4 min to 16.75 min in the Q1 scan acquisition mode. Scan speed was 10,000 from 45 m/z to 500 m/z. Aroma volatile compounds were identified by comparison with their retention indices relative to the multiple n-alkanes and by the mass spectra fragmentation pattern of each component compared to published mass spectra [50]. Three major compounds were quantified for QTL analysis: estragole, linalool and eucalyptol. The concentration of each compound was calculated as peak area. The final peak area value for each of the plants was determined by averaging the replicates by compound in absorbance units (A) and dividing by the mass (g) of the plant material.

## 2.4 Statistics

The Shapiro-Wilk Normality Test was used to test the normality of phenotypic data in R/stats v4.2.1 and skewness was assessed using R/moments v0.14.1 [51]. Kruskal-Wallis Rank Sum Test one-way analysis of variance was used as a non-parametric assessment of one-way analysis of variance to assess the global differences in phenotypic traits at a 5% level of significance ($p$-value $< 0.05$) among $F_2$ lines for each year. Spearman's Rank correlation coefficient was used to determine the strength of the linear relationship between pairs of phenotypes in R/stats v4.2.1. Phenotype data were transformed for QTL identification using Tukey's Ladder of Powers in R/rcompanion v2.4.16 [51, 52].

## 2.5 Flow cytometric analysis

Nuclear DNA content was determined with flow cytometry for the parents, hybrid and three randomly selected progeny (OCB-010, OCB-065 and OCB-180) by Heavenly Gardens (Galloway, OH, USA). Nuclei were extracted from fresh leaf tissue and nuclear DNA was stained with the CyStain® PI Absolute P reagent kit. Samples were analyzed on a BD Accuri™ C6 Plus Personal Flow Cytometer (BD Biosciences, Franklin Lakes, NJ, USA). Laser excitation was set to 488 nm and 640 nm. Emissions were detected through 4 colors with standard optical filters: FL1 533/30 nm, FL2 585/40 nm, FL3 > 670 nm and FL4 675/25 mm. Tomato (*Solanum lycopersicum*) and Daylily 'Purple Pixie Gumdrop' (*Hemerocallis*) with estimated genome sizes (2C DNA content) of 2.05 Gb and 8.59 Gb, respectively, were included as internal controls for each sample.

## 2.6 DNA extraction

Young leaf tissue was frozen in liquid nitrogen and ground with a bead mill (TissueLyser II; Qiagen, Hilden, Germany) using tungsten carbide beads and adapter sets pre-cooled at -80˚C. Genomic DNA (gDNA) was extracted using the E.Z.N.A.® SP Plant DNA Kit (Omega Bio-Tek, Norcross, GA) and assessed for quantity and quality using 260/280 absorbance ratios on a spectrophotometer (Nanodrop, Thermo Fisher Scientific, Waltham, MA). Samples of high quality (A260/280 ratio between 1.8 and 2.0) were diluted to 50 ng/μL for DNA sequencing library preparation.

## 2.7 Library preparation

Double digest restriction-site associated DNA sequencing (ddRADseq) libraries were prepared with the enzymes *PstI* (NEB, USA) and *MspI* (NEB, USA) according to the methods of Pyne et al. [16], which was modified from the original approach of Poland et al. [53]. One library was prepared for each progeny and at least ten libraries were prepared for each parent to generate a 10x parental sequencing depth. Libraries were quantified using a Qubit fluorometer (Thermo Fisher Scientific, Waltham, MA). Sample libraries were normalized to 6–7 ng/uL and pooled into 5 sets with 48 individual libraries per set. Each pooled library set was sequenced in its own lane of a Hi-Seq 2000 (Illumina, USA) using 2x150 bp, paired-end sequencing by Genewiz (Azenta Life Sciences, South Plainfield, New Jersey).

## 2.8 SNP calling

The Stacks v2.60 [54] software was used to convert raw paired-end reads into genotype data with BWA v0.7.17 [55] alignment and SAMtools v1.8 [56] file processing. Sets were filtered for quality, trimmed to 100 bp and demultiplexed with the Stacks process radtags program. Reads with an uncalled base or low-quality scores were removed. Replicate parent radtag files were

concatenated. The *Ocimum basilicum* reference genome [20] was indexed with the BWA index command to which progeny and parent radtags were aligned with the BWA mem command. Shorter split hits were marked as secondary. Resulting SAM files were converted to BAM files and sorted with the SAMtools view and sort commands. Loci were built using the Stacks ref_map.pl using program default settings. The Stacks populations program was used to export loci as an $F_2$ mapping population in the R/QTL format with a minimum percentage of progeny individuals required to process a locus set to 90%.

## 2.9 Genetic map construction

Linkage mapping was performed using R/QTL v1.52 [57] in R/RStudio v4.2.1 [51, 58]. Null, duplicate and potentially switched markers were removed. The general likelihood ratio was used to estimate the genetic map with the markerlrt command instead of recombination fractions with the est.rf command due to extensive segregation distortion. Segregation distortion was determined with a Bonferroni correction (*p*-value < 0.01). Markers aligning to LG99 were moved to linked groups or dropped if there was no linkage. Problematic markers were identified with the droponemarker command and markers with no linkage were eliminated from the analysis. One individual was removed as an outlier due to its high number of crossover events. The genotyping error rate was calculated with a log10 likelihood estimate and used to identify and subsequently remove genotyping errors.

## 2.10 QTL analysis

QTL mapping was conducted in R/QTL. The hidden Markov model calculated the probabilities underlying genotypes with the calc.genoprob command at a maximum distance of 1 cM and a genotyping error rate of 0.0125. Significance thresholds ($\alpha$ = 0.05) were determined for the single-QTL model using the scanone command with 1,000 permutations for each replicate. A genome scan with a single-QTL model was performed using the Haley-Knott regression mapping method and 1,000 permutations. QTLs were deemed statistically significant when the LOD exceeded the significance threshold. Confidence intervals were determined with 95% bayes credible intervals. QTL effect size was estimated from linear regression using the effectscan command. A genome scan with a two-QTL model was also performed using the Haley-Knott method and 1,000 permutations. A maximum distance of 2.5 cM was used to calculate the probabilities underlying genotypes with a genotyping error rate of 0.0125. Percent variance explained (PVE) was calculated per phenotype as PVE = $1-10^{(-(2/n)*LOD)}$ where n is the sample size (n = 199). Significance thresholds ($\alpha$ = 0.05) were determined for the 2D model using scantwo with 1,000 permutations for each replicate. MQM was implemented using the fitqtl command based on the significant results of the single and 2D analyses. The commands addint, addqtl and addpair were used to search for interactions and QTLs outside of the model. The model was further tested with forward/backward selection via penalized LOD scores using stepwiseqtl.

## 2.11 Gene annotation

The MAKER pipeline was used to create a draft annotation of the *O. basilicum* 'RUSB22' sweet basil genome. Parts of the annotation were completed on the Jetstream Cloud Atmosphere courtesy of XSEDE [59–61]. A repeat library specific to sweet basil was generated using RepeatMasker [62] and MITE-Hunter [63]. The 'RUSB22' trinity assembly from Allen et al. [64] and the proteome from *Arabidopsis thaliana* and *Thymus vulgaris* [65] along with the repeat library served as the input fasta files for the maker opts control file. Initial MAKER annotations were generated based on the aligned transcript and protein sequence alignment.

MAKER gene models with AED of 0.25 or better and greater than 50 amino acids in length were selected for SNAP training [66]. This first round of *ab initio* gene prediction by SNAP was used to improve upon the next round of MAKER annotation and each round of *ab initio* gene prediction was used to train subsequent rounds. MAKER annotation was run iteratively for three rounds of SNAP training. The predicted transcripts were provided as input into the software package GOFeat [67, 68]. GOFeat aligns transcript sequences to several databases including NCBI, UniProt, InterPro, KEGG, Pfam and SEED.

Flanking sequences for SNPs generated from the OCB mapping population that were associated with chilling tolerance and aroma QTLs were aligned to the annotated 'RUSB22' genome using BLAST and the Burrow-Wheeler Aligner [69]. Annotations within the QTLs for chilling response and aroma compounds were extracted (S1 Table). Annotations in the QTLs associated with cold response were searched for key words including cold, stress, temperature, chill, freeze and freezing and specific gene or gene families. Annotations in the QTLs associated with aroma were searched for key words, including aroma, smell, taste, flavor, sensor and volatile and specific gene or gene families.

## 3. Results

### 3.1 Chilling assay

The chilling response of the 2 parents, $F_1$ hybrid and 200 $F_2$ individuals were assessed in October 2021, December 2021 and February 2022. In all evaluations, the parents represented the extreme phenotypic classes (Fig 1A). In each of the evaluations, the chilling tolerant parent 'CB15' consistently exhibited <5% necrosis. In October and February, the chilling sensitive parent 'RU Obsession DMR' exhibited >50% necrosis. In December, 'RU Obsession DMR' exhibited only ~9% necrosis, however the average chilling response was lower in this replicate than the others and the 'RU Obsession DMR' parent still represented the extreme sensitive phenotype. The average percent necrosis of the entire population was 21.3% in October, 6.8% in December and 18.2% in February (Fig 1B).

After comparing each of these three assessments, 19% of the $F_2$ individuals displayed a "chilling tolerant" response similar to 'CB15' (<5% necrosis), 12% displayed a "chilling

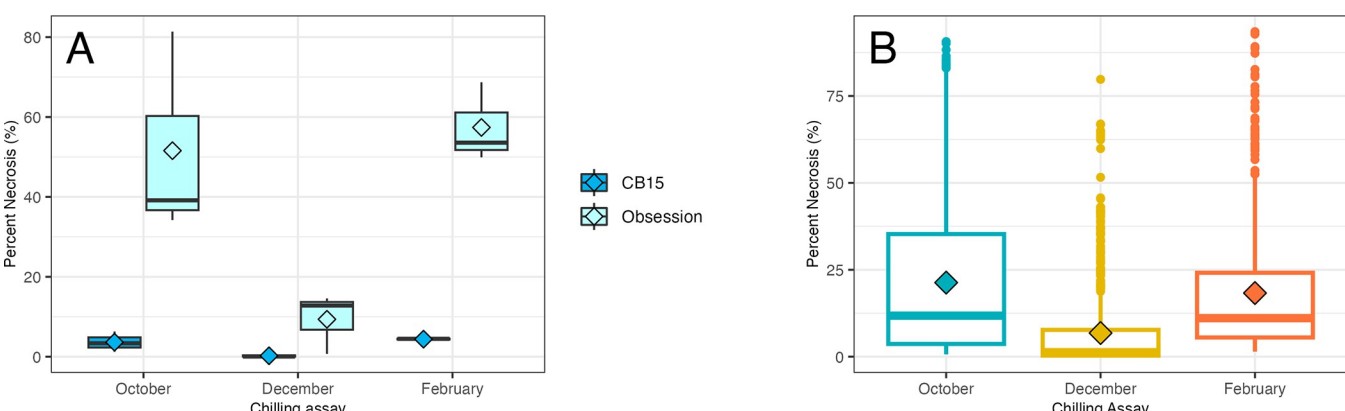

**Fig 1.** Response of the sweet basil mapping population parents (A) and $F_2$ individuals (B) to chilling temperatures measured by percent necrosis. A. Parents 'CB15' (chilling tolerant) and 'RU Obsession DMR' (chilling sensitive) represented the extreme phenotypic responses across all replicates. The December replicate showed a milder segregation of chilling response but still highlighted the extremes exhibited by the parents. B. $F_2$ individuals experienced the strongest chilling response in the October assay (average necrosis: 21.3%), followed by the February assay (18.2%), and least in the December assay (6.8%). A higher percent necrosis (100%) indicates increased chilling sensitivity, while a lower value (0%) signifies chilling tolerance. The diamond symbolizes the mean, while the central lines denote the median.

sensitive" response similar to the 'RU Obsession DMR' parent (>30% necrosis) and 69% displayed a "median response" similar to the $F_1$ (5–30% necrosis) as shown in Fig 2. The phenotype segregated quantitatively in the mapping population with a continuous variation of chilling response. The frequency of distributions was non-normal ($p$-value < 0.05) and skewed towards tolerance (0.98–2.4), thus the data were transformed for analysis.

### 3.2 Aroma volatile analysis

The aroma profiles of the two parents, $F_1$ hybrid and 200 $F_2$ individuals were analyzed for each plant in triplicate. Three major aromatic compounds were detected in the parent with the potential for segregation: estragole ($C_{10}H_{12}O$), eucalyptol ($C_{10}H_{18}O$), and linalool ($C_{10}H_{18}O$). The volatile profile of 'CB15' exhibited high estragole (3.48E+08 A/g), low eucalyptol (1.74E+08 A/g) and low linalool (1.72E+08 A/g) as shown in Fig 3A. In contrast, the volatile profile of 'RU Obsession DMR' exhibited low estragole (0 A/g), high eucalyptol (2.82E+08 A/g) and high linalool (3.09E+08 A/g).

The phenotypes of the aroma compounds segregated quantitatively in the mapping population with a continuous variation for the aroma compounds. Estragole had the highest mean peak area, followed by linalool and eucalyptol in the $F_2$ population (Fig 3B). Despite having the highest mean peak area, a large portion of the $F_2$ population (41.5%) had no detectable levels of estragole and the rest had moderate to high levels of estragole (7.6e+07–1.44e+09 A). All $F_2$ individuals had moderate to high levels of linalool (1.3e+07–8.2e+08 A) and low to high levels of eucalyptol (2.3e+05–4.0e+08 A). Transgressive segregation was also observed with 42% of $F_2$ individuals having higher estragole levels than the 'CB15' parent, 61% having lower levels of eucalyptol content than the 'CB15' parent, 30% having lower linalool levels than the 'CB15' parent and 42% having higher linalool levels than the 'RU Obsession DMR' parent. The

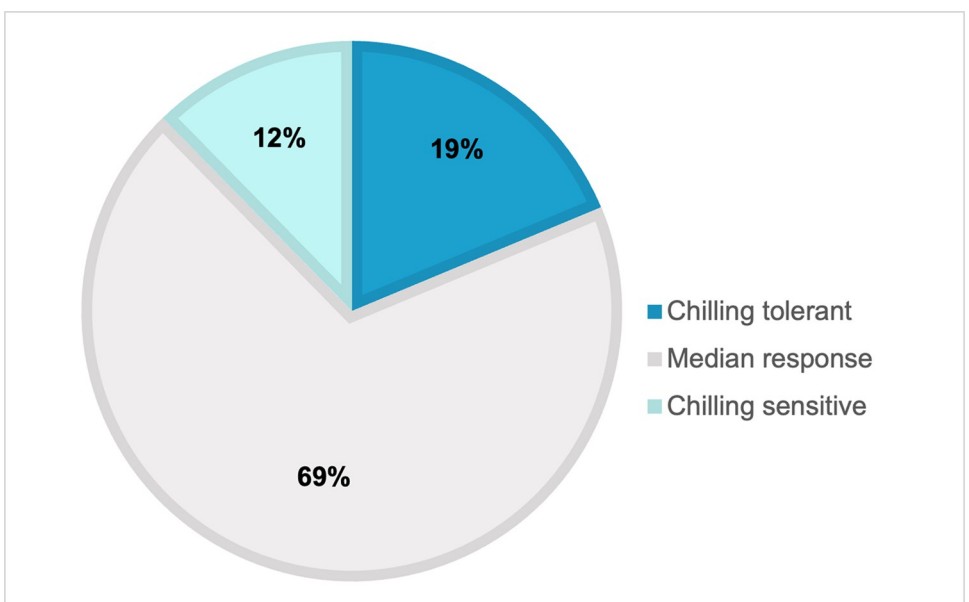

**Fig 2. Categorization of the $F_2$ mapping population's response to chilling temperatures.** Chilling tolerant $F_2$ basil plants, mirroring the 'CB15' parent, exhibited <5% necrosis while chilling sensitive $F_2$ plants, mirroring the 'RU Obsession DMR' parent, exhibited >30% necrosis after all three chilling assays. The $F_2$ plants with a median response, comparable to the $F_1$ hybrid, exhibited 5–30% necrosis. Most $F_2$ individuals displayed a median response (69%), 19% displayed a chilling tolerant response and 12% exhibited a chilling sensitive response.

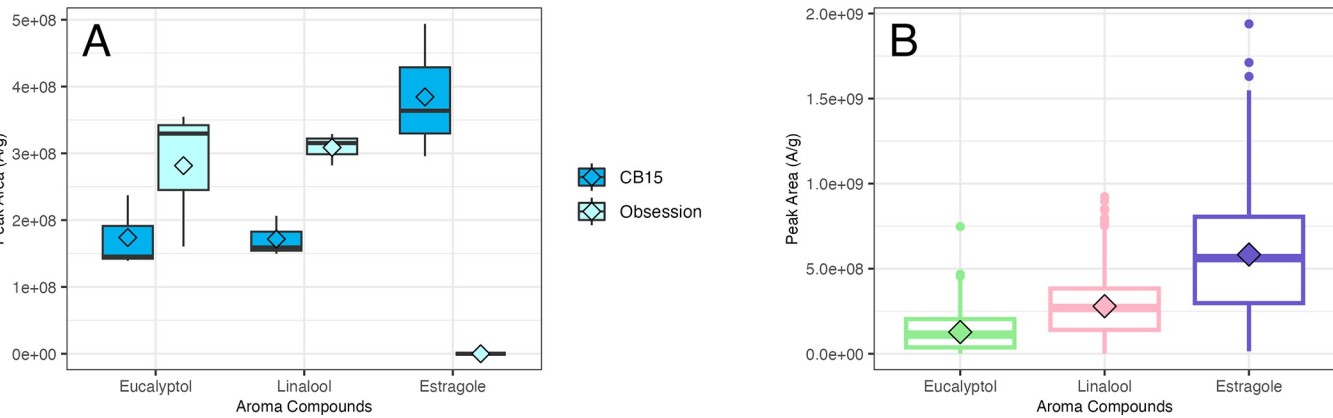

**Fig 3.** Aroma compound levels of the sweet basil mapping population parents (A) and $F_2$ individuals (B). A. In all replicates, the parents exhibited distinct aromatic profiles: 'CB15' consistently showed high estragole but low eucalyptol and linalool concentrations, while 'RU Obsession DMR' was characterized by its low estragole but high eucalyptol and linalool content. B. Among the $F_2$ population, estragole emerged with the highest mean peak area, followed by linalool and eucalyptol. A higher peak area (A/g) indicates higher concentrations of the aroma compound per gram of plant material and the lower peak area indicates lower concentrations. The diamond represents the mean, the central lines represent the median.

frequency of distributions was non-normal ($p$-value $< 0.05$), thus the data were transformed for analysis.

### 3.3 Phenotypic correlations

Chilling assays were moderately correlated to each other ($\rho = 0.29$–$0.47$; Fig 4). There was no correlation between the results of the chilling assays and the aroma analyses ($\rho = -0.03$–$0.14$).

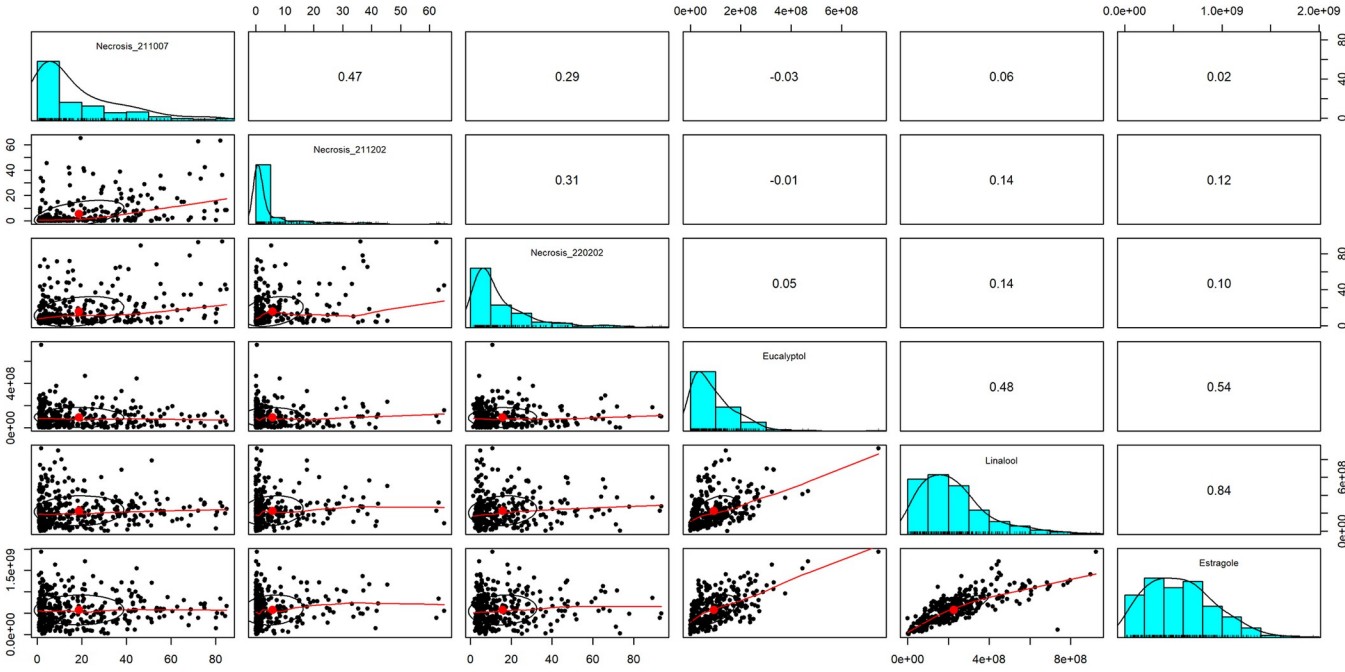

**Fig 4. Correlations between phenotypes.** Bivariate scatter plots are shown in the bottom lefthand corner, histograms on the diagonal and the Spearman correlation in the upper righthand corner. Chilling assays were moderately correlated with each other and aroma compounds were moderately to highly correlated with each other. There was no correlation between chilling response and aroma profile.

Aroma compounds were moderately to strongly correlated with each other. Eucalyptol was moderately correlated to linalool ($\rho = 0.48$) and estragole ($\rho = 0.54$). Linalool was highly correlated to estragole ($\rho = 0.84$).

### 3.4 Flow cytometric analysis

The total estimated genome size of all analyzed accession was 4.85–4.88 Gb (2C DNA content). The similarity in DNA content strongly suggests that the parents, resulting hybrid and $F_2$ progeny each have the same number of chromosomes (S1 Fig). This genome size is consistent with previous literature for tetraploid basil [9, 10].

### 3.5 Genotyping and genetic mapping

The 200 $F_2$ progeny and parents were genotyped using ddRADseq. Sequencing yielded 1.98 billion raw reads and 593,628 Mbases. The mean quality score was 35.73. Reads retained from the parents were 11-27x that of the $F_2$ individuals. The Stacks software pipeline was used to generate 67,575 loci with 3,558 mappable markers. Removal of null, duplicate, potentially switched and unlinked markers and problematic individuals resulted in a genetic map of 1,761 polymorphic SNP markers ordering into 25 linkage groups (Figs 5 and 6). The original reference map has 26 linkage groups, however LG21 is absent in this analysis due to an insufficient number of quality markers. The total distance on the genetic map was 2,241.8 cM, with an average linkage group length of 90 cM. The shortest LG was 5 cM, while the longest LG was 162 cM. The average number of markers per LG was 70.4, varying from 13 to 225. Segregation of genotypes was distorted with a ratio of 28.4% AA: 24.5% AB: 47.1% BB (~1:1:2) instead of 1 AA: 2 AB: 1 BB, with 98.5% of markers considered to be highly distorted ($p$-value $< 0.01$). The heatmap of the marker-pairwise estimated recombination fractions vs LOD scores supported that there is only linkage within each LG (S2 Fig).

### 3.6 QTL identification for chilling injury

Five statistically significant QTLs were detected for chilling response with one significant QTL consistent across all three replications in the single-QTL model (Fig 5; Table 1). The LOD thresholds ($\alpha = 0.05$) were determined to be 3.70, 3.96 and 3.81 for the chilling assays conducted in October, December and February, respectively. The QTL on LG24 (*qCH24)* was significant across all three replicates (LOD = 12.8, 10.4 and 4.1) and had the largest effect in the October and December replicates (PVE = 25.6% and 21.4%), with the second largest effect in the February replicate (PVE = 9.0%). In all replicates, the confidence interval for *qCH24* started at 0.3 cM and spanned to ~70 cM. The QTL on LG16 (*qCH16*) was only significant in the February replicate (LOD = 4.3) and had a slightly higher effect (PVE = 9.5%) than *qCH24*. The QTL on LG7 (*qCH7*) was significant in the October and December replicates (LOD = 4.7 and 6.1; PVE = 10.3% and 13.2%). The QTL on LG13 (*qCH13*) was significant in the October replicate only (LOD = 4.5; PVE = 9.8%) and the QTL on LG23 (*qCH23*) was significant in the February replicate only (LOD = 4.0; PVE = 8.8%).

Two additional loci were detected in some of the replicates in the two-dimensional (2D)-QTL model and the multiple-QTL model (MQM). A QTL on LG1 was significant in all replicates at 28.1–48.1 cM and a QTL on LG14 at 19.6 cM interacting with LG1 was significant in the February replicate. All loci from the single-QTL model were significant except LG7 in the October replicate. MQM of the October replicate supported a single-QTL model (LOD = 21.5), however the December and February replicates supported a four-QTL (LOD = 23.5) and five-QTL model with one interaction (LOD = 27.2), respectively. Results of

**October**

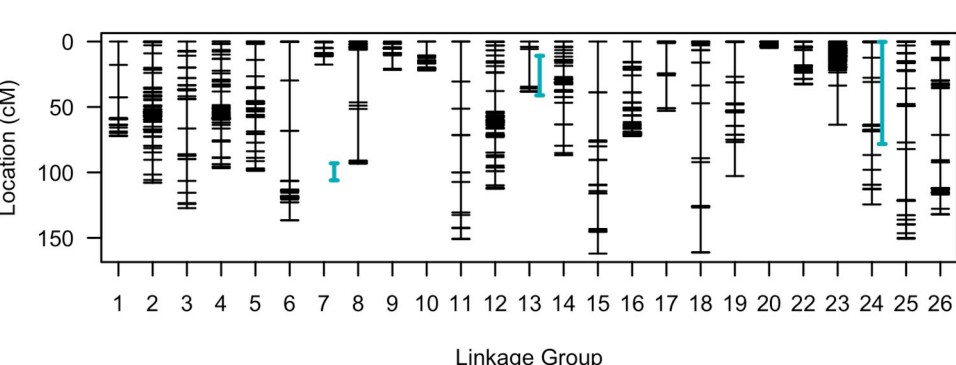

**December**

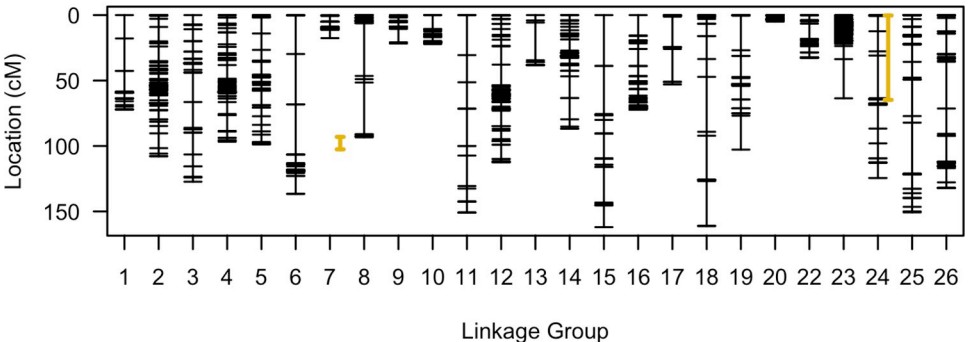

**February**

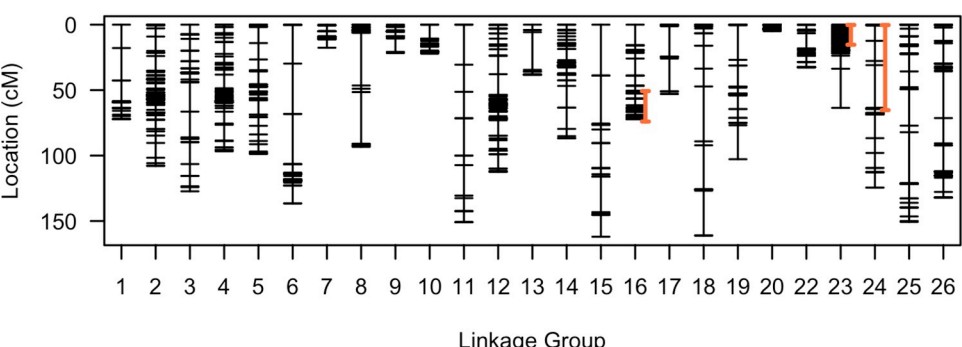

**Fig 5. Genetic map with confidence intervals for significant QTLs in each chilling assay.** A large effect QTL was located on LG24 in all three chilling assays spanning from 0.3–78 cM. While other significant QTLs were noted on LG7, LG13, LG16 and LG24, their presence was not consistent across all experiments. Confidence intervals were determined with 95% Bayesian credible intervals without expanding to the nearest markers.

the MQM analysis across all three replicates in time indicate that the QTL on LG24 represents the major QTL for chilling response in this population.

Marker 938232 was estimated as the marker closest to the peak of *qCH24*. Analysis of the phenotype effects of the markers associated with this QTL indicated that $F_2$ individuals with

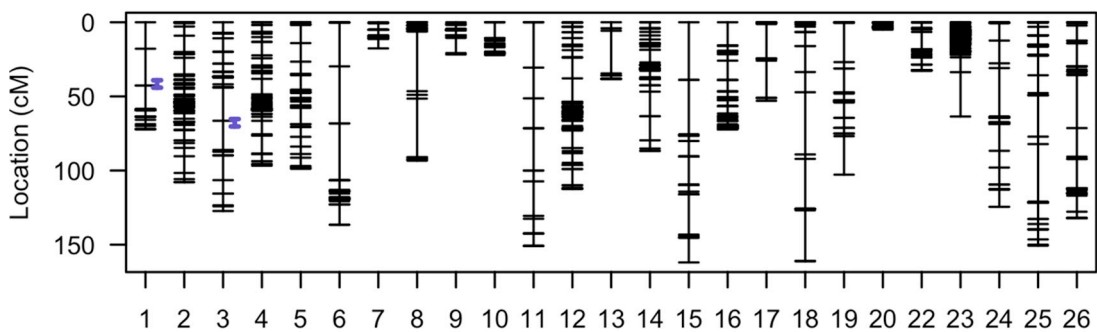

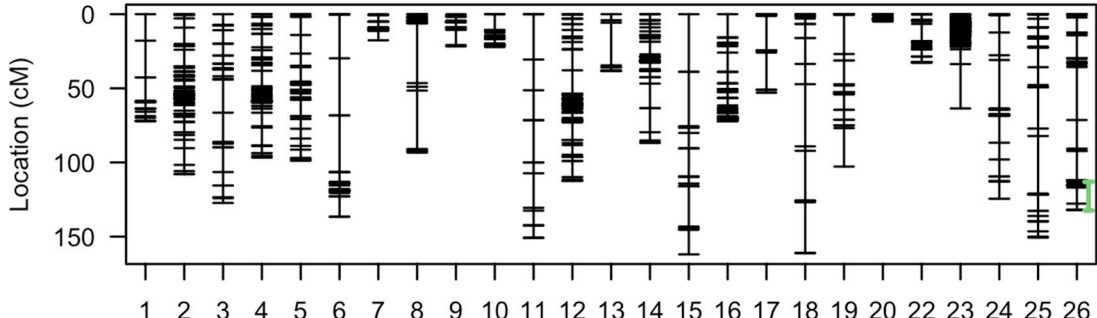

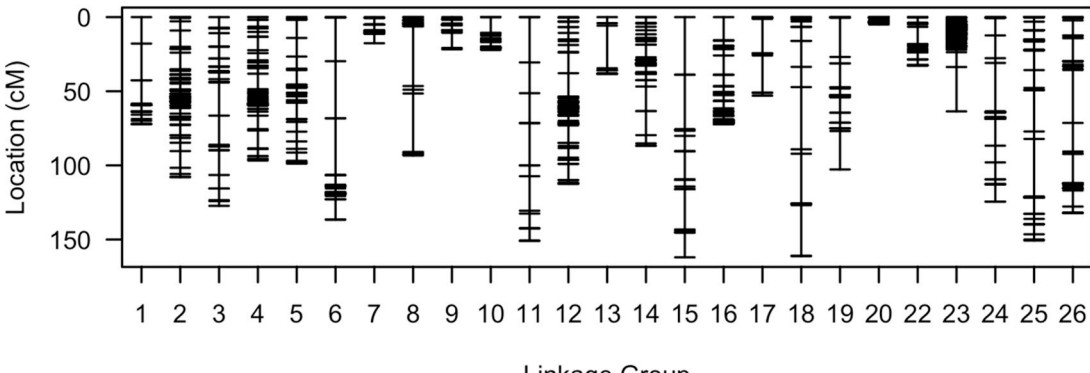

**Fig 6. Genetic map with confidence intervals for significant QTLs for each aroma volatile compound.** Two QTLs were detected for estragole (purple) spanning from 18.3–19.2 cM on LG1 and 65.2–70.3 cM on LG3 in the single-QTL model. A QTL was detected for eucalyptol (green) spanning from 113.0–132.4 cM on LG26. No QTLs were detected for linalool. Confidence intervals were determined with 95% Bayesian credible intervals without expanding to the nearest markers.

**Table 1. Loci where LOD scores exceeded the LOD threshold (α = 0.05) in the single-QTL model.**

| Phenotype | Replicate | QTL | LG | Peak position | | | Closest marker | | | Interval (cM) |
|---|---|---|---|---|---|---|---|---|---|---|
| | | | | cM | LOD | PVE (%) | Marker | cM | LOD | |
| Chilling | October | qCH7 | 7 | 98.1 | 4.7 | 10.3 | 302350 | 98.1 | 4.7 | 93.0–106.0 |
| | December | | | 98.1 | 6.1 | 13.2 | 302350 | 98.1 | 6.1 | 93.0–102.5 |
| Chilling | October | qCH13 | 13 | 38.8 | 4.5 | 9.8 | 589396 | 38.8 | 4.5 | 10.9–41.1 |
| Chilling | February | qCH16 | 16 | 63.2 | 4.3 | 9.5 | 699095 | 63.2 | 4.3 | 51.7–74.0 |
| Chilling | February | qCH23 | 23 | 9.8 | 4.0 | 8.8 | 923381 | 9.8 | 4.0 | 0.2–15.3 |
| Chilling | October | qCH24 | 24 | 67.9 | 12.8 | 25.6 | 938232 | 67.9 | 12.8 | 0.3–78.3 |
| | December | | | 57.3 | 10.4 | 21.4 | 936090 | 64.8 | 10.0 | 0.3–64.8 |
| | February | | | 40.3 | 4.1 | 9.0 | 930202 | 0.3 | 3.6 | 0.3–65.3 |
| Estragole | NA | qEST1 | 1 | 42.7 | 19.2 | 35.9 | 765 | 42.7 | 19.2 | 39.1–44.1 |
| Estragole | NA | qEST3 | 3 | 68.7 | 12.5 | 25.2 | 120144 | 68.7 | 12.5 | 65.3–70.3 |
| Eucalyptol | NA | qEUC26 | 26 | 130.0 | 8.0 | 16.9 | 989820 | 132.4 | 7.6 | 113.0–132.4 |

Peak position was determined from a genome scan with a single-QTL model using the Haley-Knott regression mapping method and 1,000 permutations. Closest markers and confidence intervals were determined with 95% bayes credible intervals. Peak position (cM) and markers represent those with the highest LOD). The LOD thresholds (α = 0.05) were determined to be 3.70, 3.96 and 3.81 for the chilling assay replicates in October, December and February, respectively, 3.80 for estragole and 3.66 for eucalyptol.

two alleles from 'CB15' (AA) were the most chilling tolerant while those with one allele from each parent (AB) were the most sensitive as exemplified by the marker effect plots for marker 938232 (Fig 7). The other markers linked to *qCH24* followed the same pattern, suggesting non-additive or dominant gene action for chilling sensitivity. Marker 930202 at 0.3 cM was estimated to be the QTL end-point in all three chilling assays. Marker 946237 at position 87.0

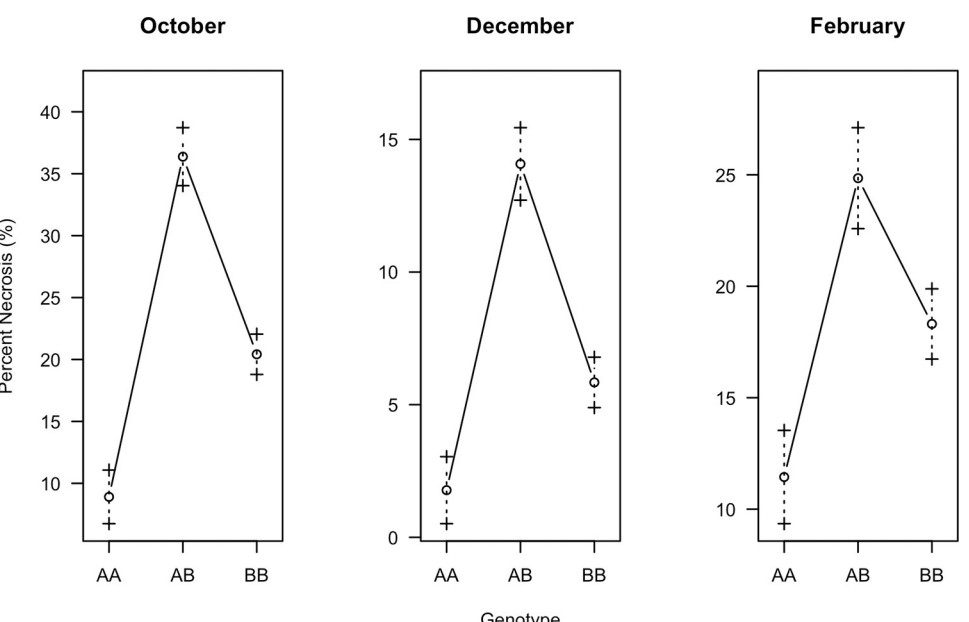

**Fig 7. Effect plots of marker 938232 in each chilling assay.** Marker 938232 was the closest marker to the QTL peak on LG24 and suggests non-additive gene action where the chilling sensitive genotype (BB) is dominant. The AA genotype represents F₂ individuals that inherited two alleles from the 'CB15', the BB genotype represents F₂ individuals that inherited two alleles from the 'RU Obsession DMR' parent and the AB represents F₂ individuals that inherited one allele from each.

cM was estimated to be the nearest flanking marker of the other end-point in the October and February assay while marker 936090 at 65 cM was estimated to be the other end point in the December assay.

### 3.7 QTL identification for aroma compounds

Three significant QTLs were detected for aroma compounds in the single-QTL model (Fig 6; Table 1). QTLs were detected for estragole on LG1 (*qEST1*; LOD = 19.2; PVE = 35.9%) and LG3 (*qEST3*; LOD = 12.5; PVE = 25.2%) with *qEST1* having a larger effect. A QTL was detected on LG26 (LOD = 8.0; PVE = 16.9%) for eucalyptol. No significant QTLs were detected for linalool. The LOD thresholds ($\alpha$ = 0.05) were determined to be 3.80, 3.66 and 3.78 for estragole, eucalyptol and linalool, respectively. No additional loci were detected as interacting with the significant loci in the 2D-QTL model or MQM. Furthermore, MQM refined that only the QTL on LG1 and not that on LG3 was significant for estragole.

Marker 765 was estimated as the closest marker to the peak of *qEST1* (Fig 8). Analysis of the phenotype effects of the markers associated with this QTL indicated that $F_2$ individuals with two alleles from 'CB15' (AA) had the highest levels of estragole while those with one allele from each parent (AB) had the lowest levels of estragole. The other markers linked to *qEST1* observed the same trend, suggesting non-additive gene action. Marker 206 at 17.9 cM and marker 5502 at 58.6 cM were estimated to be the nearest flanking markers to the approximate end points of the QTL.

Marker 989212 was estimated as the closest physical marker to the peak of *qEUC26*. The gene action is less clear for this QTL as the phenotypic effects were similar for the $F_2$ inheriting the AA genotype and the BB genotype. Marker 978874 at 113.0 cM and marker 989820 at 132.4 were estimated to be the nearest flanking markers to the approximate end points of the QTL.

### 3.8 Gene annotation

Several putative genes within the QTLs associated with cold response (*qCH7*, *qCH13*, *qCH16*, *qCH23* and *qCH24*) were annotated with genes homologous to cold response and stress proteins in general (Table 2). Two annotations with functional responses to cold temperature were identified, including cold shock protein 1 on LG7 and putative transcriptional regulator UXT on LG23. Late embryogenesis abundant (LEA) proteins, 14-3-3 proteins, abscisic acid (ABA) related isoforms and binding factors, fatty acid desaturases (FADs) and proline related enzymes were annotated to the cold responsive QTLs and of particular interest for their relationship to cold stress [70–74]. Other annotations of interest included 11 universal or general stress-response proteins on LG7, LG13, LG16, LG23 and LG24. Many putative genes on the QTLs associated with cold response were associated with regulator proteins including 512 annotations for kinases and 159 annotations for transcription factors (S1 Table). Many annotations were also associated with other related to cold response including 190 transport, 126 membrane and 33 heat shock proteins.

Putative genes that are homologous to terpene enzymatic proteins were identified on the QTLs associated with aroma compounds (*qEST1* and *qEUC26*) as shown in Table 3. One gene annotated to LG1 as caffeic acid O-methyltransferase, which is a critical enzyme in the phenylpropanoid metabolic pathway [75]. Caffeoyl-CoA O-methyltransferase is also an important enzyme in phenylpropene synthesis for which 6 annotations were identified on LG1 and LG26 [76]. Annotations for 2 terpene synthases and 1 terpene cyclase/mutase family member were identified on LG26, which comprised a significant QTL for the monoterpene eucalyptol.

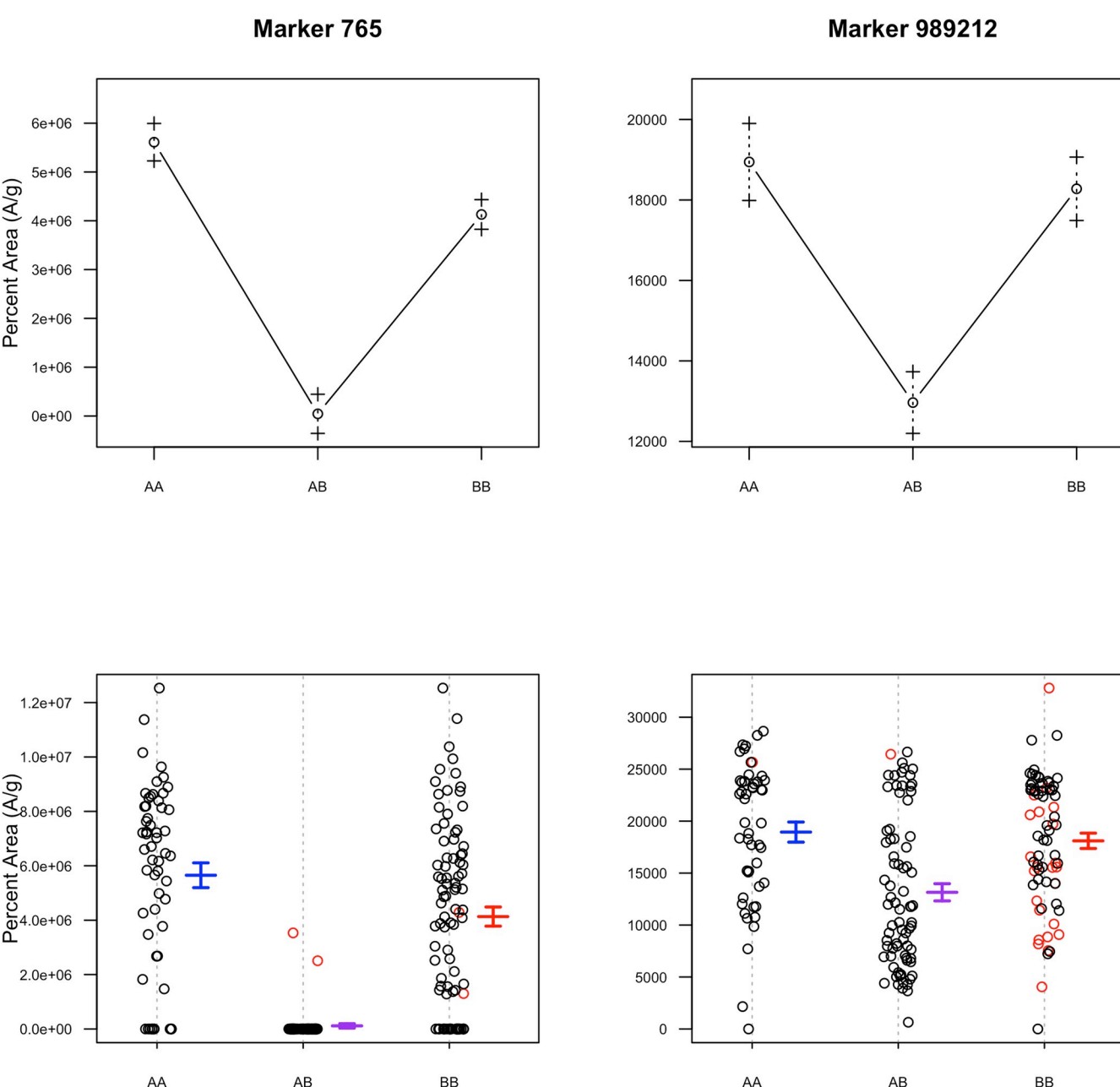

**Fig 8. Effect plots of markers on QTL for estragole and eucalyptol.** Marker 765 was the closest marker to the QTL peak on LG1 for estragole. Marker 765 suggests non-additive gene action where the low estragole genotype (BB) is dominant. Marker 989212 was the closest marker to the QTL peak on LG26 for eucalyptol. Marker 989212 is less supportive. The AA genotype represents F₂ individuals that inherited two alleles from the 'CB15', the BB genotype represents F₂ individuals that inherited two alleles from the 'RU Obsession DMR' parent and the AB represents F₂ individuals that inherited one allele from each.

## 4. Discussion

Basil is known to be one of the most chilling sensitive horticultural crops with temperatures less than 10°C (50°F) inducing severe if not total injury of basil leaves [38–40]. Chilling tolerance was identified in breeding line 'CB15' and was observed to be a heritable trait in the F₂ progeny (Figs 1 and 2). Chilling tolerance was defined as <5% necrosis in the cut leaves after 4

**Table 2. Genome annotations of interest within QTLs associated with cold response.**

| LG | Homologous gene | Function | Species | AED | Uniprot Accession |
|----|-----------------|----------|---------|-----|-------------------|
| 7 | Cold shock protein 1 | NA | *Striga asiatica* | 0.07 | A0A5A7RB26 |
| 7 | Universal stress protein 27 | NA | *Salvia miltiorrhiza* | 0.04 | A0A290YXU3 |
| 7 | Stress-response A/B barrel domain-containing protein | NA | *Handroanthus impetiginosus* | 0.39 | A0A2G9I0I6 |
| 7 | Stress response protein nst1 | NA | *Sesamum indicum* | 0.27 | A0A6I9UJN7 |
| 7 | LEA_2 domain-containing protein | NA | *Salvia splendens* | 0.26 | A0A4D8ZMQ8 |
| 7 | 14-3-3-like protein 16R | NA | *Sesamum indicum* | 0.04 | A0A6I9U980 |
| 7 | Protein ABA DEFICIENT 4 chloroplastic-like isoform | NA | *Sesamum indicum* | 0.02 | A0A6I9U8G4 |
| 7 | ABA responsive element binding factor | NA | *Salvia splendens* | 0.35 | A0A4D9BA86 |
| 7 | Proline dehydrogenase | NA | *Salvia splendens* | 0.07 | A0A4D8ZXX4 |
| 7 | Omega-3 fatty acid desaturase | NA | *Perilla frutescens* | 0.00 | O04807 |
| 7 | FAD-binding FR-type domain-containing protein | NA | *Vitis vinifera* | 0.29 | E0CUW8 |
| 7 | FAD-binding FR-type domain-containing protein | NA | *Salvia splendens* | 0.06 | A0A4D9AH61 |
| 7 | FAD-binding FR-type domain-containing | NA | *Salvia splendens* | 0.38 | A0A4D9AP55 |
| 13 | Universal stress protein 23 | NA | NA | 0.25 | A0A290YXT7 |
| 13 | Proline iminopeptidase | NA | *Handroanthus impetiginosus* | 0.19 | A0A2G9GH38 |
| 13 | Omega-3 fatty acid desaturase | NA | *Perilla frutescens* | 0.25 | A0A2H4MCG8 |
| 16 | Universal stress protein 7 | protein kinase activity; ATP binding; protein phosphorylation | NA | 0.03 | A0A290YXQ1 |
| 16 | Universal stress protein 22 | NA | *Salvia miltiorrhiza* | 0.02 | A0A290YXT6 |
| 16 | Universal stress protein 19 | NA | *Salvia miltiorrhiza* | 0.02 | A0A290YXS5 |
| 16 | Universal stress protein 21 | NA | *Salvia miltiorrhiza* | 0.01 | A0A290YXU1 |
| 16 | Universal stress protein 8 | NA | *Salvia miltiorrhiza* | 0.14 | A0A290YXN9 |
| 16 | LEA_2 domain-containing protein | NA | *Erythranthe guttata* | 0.17 | A0A022PQW9 |
| 16 | LEA_2 domain-containing protein | NA | *Salvia splendens* | 0.02 | A0A4D8YFU8 |
| 16 | ABA-responsive element ABRE-binding transcription factor1 | NA | *Salvia miltiorrhiza* | 0.40 | A0A2U9Q8N8 |
| 16 | FAD synthase isoform X1 | NA | *Perilla frutescens* | 0.11 | A0A6I9T1J8 |
| 23 | Putative transcriptional regulator UXT | protein folding; response to cold | NA | 0.16 | A0A2G9GFB7 |
| 23 | Universal stress protein 13 | NA | *Salvia miltiorrhiza* | 0.12 | A0A290YXS7 |
| 23 | 14-3-3 protein 9 | NA | *Sesamum indicum* | 0.38 | A0A6I9UP33 |
| 23 | ABA responsive element binding factor | NA | *Salvia splendens* | 0.18 | A0A4D9BM39 |
| 24 | Universal stress protein 13 | NA | *Salvia miltiorrhiza* | 0.08 | A0A290YXS7 |
| 24 | 14-3-3 protein 9 | NA | *Sesamum indicum* | 0.38 | A0A6I9UP33 |
| 24 | ABA responsive element binding factor | NA | *Salvia splendens* | 0.17 | A0A4D9BM39 |
| 24 | Omega-3 fatty acid desaturase | NA | *Perilla frutescens* | 0.11 | A0A2H4MEV2 |

Putative genes of interest within significant QTL were those homologous to genes associated with cold or stress. NA indicates information not available. LG = linkage group.

days at 3–5°C, while chilling sensitivity was defined as >30% necrosis in the same conditions (Fig 2). These delineations were selected to reflect the response of the chilling tolerant and chilling susceptible parents. As a result, only 19% of the $F_2$ population had a chilling tolerant response while 88% had a median or highly chilling sensitive response. Thus, the gene action is likely non-additive or dominant for chilling sensitivity, which is also supported by the marker effects within *qCH24* (Fig 7).

**Table 3. Genome annotations of interest within QTLs associated with aroma.**

| LG | Homologous gene | QTL Association | Species | AED | Uniprot Accession |
|----|----|----|----|----|----|
| 1 | Caffeic acid O-methyltransferase | Estragole | *Catalpa bungei* | 0.05 | A0A411ASI0 |
| 1 | Caffeoyl-CoA O-methyltransferase | Estragole | *Salvia splendens* | 0.17 | A0A4D9BQK0 |
| 1 | Caffeoyl-CoA O-methyltransferase | Estragole | *Salvia dorisiana* | 0.45 | A0A6G5RUA9 |
| 26 | Terpene synthase | Eucalyptol | *Prunella vulgaris* | 0.08 | A0A6B7L8X6 |
| 26 | Terpene synthase | Eucalyptol | *Scutellaria barbata* | 0.25 | A0A6B7LU35 |
| 26 | Terpene cyclase/mutase family member | Eucalyptol | *Ocimum basilicum* | 0.03 | A0A0D3L308 |
| 26 | Caffeoyl-CoA O-methyltransferase | Eucalyptol | *Erythranthe guttata* | 0.23 | A0A022QTM1 |
| 26 | Caffeoyl-CoA O-methyltransferase | Eucalyptol | *Salvia splendens* | 0.46 | A0A4D9ASU9 |
| 26 | Caffeoyl-CoA O-methyltransferase | Eucalyptol | *Erythranthe guttata* | 0.01 | A0A022QTM1 |
| 26 | Caffeoyl-CoA O-methyltransferase | Eucalyptol | *Daucus carota subsp. Sativus* | 0.20 | A0A166C3S9 |

Putative genes of interest within significant QTL were those homologous to genes associated with terpene biochemical pathways. NA indicates information not available. LG = linkage group.

The aromatic profile of the chilling tolerant parent, 'CB15' was high in estragole and has a licorice aroma, while the chilling sensitive parent, 'RU Obsession DMR' was high in linalool and eucalyptol and has a floral sweet basil aroma with cooling spicy notes. No estragole or low levels were detected in a majority of $F_2$ individuals, suggesting that the gene action is likely non-additive or dominant for the low estragole phenotype. These results are supported by the marker effect plots (Fig 8). Whereas the gene actions for eucalyptol and linalool are less clear.

The final genetic map yielded 1,761 SNPs across 25 linkage groups, averaging 90 cM distance between markers over a total map distance of 2,241.8 cM. These results are comparable to the denovo genetic linkage map of an *O. basilicum* biparental $F_2$ population that also used the parent 'SB22' [16]. The number of linkage groups were found to be the same in these studies, however, one was dropped from this analysis due to insufficient quality marker coverage and the genetic distances between markers were reduced. The final number of mappable SNPs is significantly lower than the other known genetic linkage map for *O. basilicum* generated from aligning the reads of a 'Perrie' and 'Cardinal' $F_2$ mapping population to a contiguous draft reference genome [19]. The difference in SNP yield can be due to a variety of factors including relatively high homozygosity or recent admixture between the parents [77], stricter filtering parameters and the use of a chromosome-level reduced genome, which reduces the risk of false positives [78].

Both parents in this population ('RU Obsession DMR' and 'CB15') are known to be members of the same species (*Ocimum basilicum* L.). Hybridization produced fertile $F_1$ progeny, and selfing the hybrid produced fertile $F_2$ progeny. Accessions of the same species with no fertility issues in resulting progeny are expected to have the same number of chromosomes. The general consensus is that cultivated sweet basil (*Ocimum basilicum* L.) is an allotetraploid (2n = 4x = 48) and therefore genetic mapping and QTL study methods should assume disomic inheritance or "diploidization" [3–10]. However, the prevalence of marker segregation distortion observed in this population challenges these postulations.

One hypothesis for the marker segregation distortion was that the parents differed in their chromosome numbers. Deviations from the proposed 2n = 4x = 48 chromosome model have been observed in *O. basilicum* accessions, especially in those wild collected, with chromosome numbers of 2n = 50 [11], 2n = 52 [5], 2n = 54 [13], 2n = 5x = 60 [3], 2n = 6x = 72 [6, 9, 11, 12] and 2n = 74 [6]. Digression from this assumption was also observed through construction of the first *de novo* QTL map of *O. basilicum* with the identification of 26 linkage groups instead

of 12 or even 24 [16]. Furthermore, *O. basilicum* parents with different chromosome numbers have been crossed and progeny found to be fertile despite the divergent genetic backgrounds [11]. The differing chromosome base numbers between accessions and fertile hybridization suggest a complex evolutionary history of polyploidy and aneuploidy. However, the similarity in nuclear DNA content suggests that the two parents have the same number of chromosomes (S1 Fig).

Despite the lack of evidence of multivalent formation in *O. basilicum* [4, 12], segmental allopolyploidy is a possible explanation for the segregation distortion because tetrasomic inheritance has not yet been explored in basil. Recent evidence found that tetraploid rose can exhibit partial preferential pairing, resulting in some genes adopting disomic inheritance and others tetrasomic inheritance [79]. In such cases of segmental allopolyploidy, assuming a disomic model would result in higher ratios of genetic distortion.

Marker segregation distortion could also arise due to a lack of recombination in the $F_2$ progeny. If the parents recently diverged and are too genetically similar, then there would be few differences in allele calls between the parents and thus a lower level of segregation in the progeny. This explanation would also support reduced recombination levels and lower than expected marker yield. Other mechanisms such as meiotic drive, gametic selection, zygotic or post-zygotic selection and cytoplasmic incompatibility are less likely due to the high fertility of the parents and hybrid. Further investigation of the *O. basilicum* genome is needed to determine the cause of marker segregation distortion in the current study.

*QCH24* was a significant QTL across all three chilling assay replicates spanning a relatively large confidence interval from 0.3 cM up to 78 cM on LG24. Four additional QTLs were detected in the single-QTL model and two in the MQM. All these QTLs were of smaller effect except *qCH16*, which had a slightly higher effect than *qCH24* in one replicate. Even though the chilling assays were only moderately correlated, they each detected the same significant QTL (Figs 4 and 5). The effect size of *qCH24* was the largest in the first replicate and smaller in the second and third replicate. This is likely due to the lower average percent necrosis observed across the entire $F_2$ population in the later replicates (Fig 1B). The population-wide difference in response to chilling temperatures across assays could be due to plant maturation, acclimation to the colder months or photoperiod.

One major QTL (*qEST1*) was detected for estragole on LG1 and for eucalyptol (*qEUC26*) on LG26. Another smaller effect QTL (*qEST3*) was detected in the single-QTL model for estragole on LG3, however, it was not significant in the MQM. The parents were the least different in their relative concentrations of linalool (Fig 3A), which explains why no QTLs were detected for this aroma compound. Estragole, eucalyptol and linalool levels in the $F_2$ population all positively correlated with each other (Fig 4). Thus, when an $F_2$ individual had a high concentration of one major aroma compound, it also had relatively higher concentrations of the other major aroma compounds. Although eucalyptol and linalool form through a different pathway than estragole, it is plausible that common signals regulate the various biosynthetic pathways for basil aroma, as observed in tea [80]. Especially since both pathways are spatially localized in basil glandular trichomes cells, which can enable direct crosstalk between pathways [81]. Future work could focus on exploring interactions between pathways or devising populations with more extreme phenotypic differences in the parents to identify more genomic regions directly or indirectly related to the aroma pathways.

Potential gene candidates were identified through genome annotation on the QTLs associated with chilling. Three putative genes on LG24, which contains the major QTL for chilling response, were annotated as proteins that could be associated with cold response, including universal stress protein 13, omega-3 FAD, 14-3-3 protein 9 and an ABA responsive element binding factor (Table 2). Desaturation of fatty acids via omega-3 FAD and other FADs have

been shown to improve cold tolerance through transcriptomic, lipidomics and transgenic studies in *Tripsacum*, rice and other plants [82–84]. Similarly, overexpression of 14-3-3 proteins has been found to improve cold tolerance and in *Arabidopsis* [74]. Cold activation of *CRPK1* phosphorylates 14-3-3 proteins and leads to the degradation of *CBF1* and *CBF3* [73]. This negatively regulates *COR* expression and freezing tolerance, though no *CBF* or *COR* genes were annotated on the QTLs associated with chilling. Conversely, applications of ABA have shown to improve plant response to cold stress [70, 71]. ABA mediates abscisic acid-responsive element (ABRE) binding factors (ABFs), which can be cold inducible [85]. Putative genes associated with cold temperature and stress were also identified on the linkage groups that contained minor QTLs associated with chilling response, i.e., LG7, LG13, LG16 and L23. These included cold shock, general stress, LEA, FADs, 14-3-3, ABA related and proline related proteins. LEA and proline related proteins are of interest for their highly conservative roles under cold stress [72, 82–84, 86].

Many gene candidates on the linkage groups of the major and minor QTLs related to cold response were also associated with regulatory genes, including 512 associated with kinases and 159 with transcription factors. Kinases play a crucial role in the regulation of cold tolerance in plants by modulating the activity of various proteins involved in cold acclimation, including phosphorylation of transcription factors, protein regulation, post-translational modification and signal transduction [87]. *QCH24* spans a large genetic distance, suggesting that multiple intra-QTL genes are related to chilling injury, and possibly co-localize to a regulatory gene [88].

Many genes related to transport (190), membrane (126) and heat shock (33) proteins were identified across these loci, which could further contribute to the complexity of cold response in basil. In the early stages of cold exposure, transport proteins are up-regulated, such as ATP binding cassettes, sugar and phosphate transporters, as well as ATPases. Over a longer duration, these transporters are down-regulated, whereas genes linked to carbohydrate metabolism are up-regulated [89]. Membrane-related genes are crucial for cold response in plants as they govern the compositional and functional adaptations of the plasma membrane that maintain cellular metabolism, ion homeostasis, and signaling processes, ultimately contributing to cold tolerance or sensitivity [90]. Additionally, chilling temperatures initiate the accumulation of misfolded proteins, which are aided by the chaperone activity of heat shock proteins in stress tolerant plants [91].

The QTLs associated with aroma compounds yielded interesting relationships to potential gene candidates. Two candidate genes were identified as homologous to terpene synthases and one as homologous to terpene cyclase/mutase family members on LG26, which contained the QTL associated with eucalyptol, a terpene (Table 3). No phenylpropenes, propenes or propenoids were identified on LG1, which contained the QTL associated with estragole, a phenylpropene. However, a gene was annotated on LG1 as caffeic acid O-methyltransferase, which is a critical enzyme in the phenylpropanoid metabolic pathway [75]. Caffeoyl-CoA O-methyltransferase is also an important enzyme in phenylpropene synthesis for which 6 annotations were identified on LG1 and LG26 [76].

Aroma was evaluated not only for its importance in breeding, but also its potential relationship to chilling tolerance. Previous research observed that when screening sweet basils for chilling tolerance, plants exhibiting chilling tolerance were more associated with licorice aroma in contrast to a more traditional sweet basil aroma [42]. In this study, there was no correlation between chilling response in the $F_2$ progeny and the major aroma compounds: estragole, eucalyptol and linalool (Fig 4). There was also no overlap in the major QTL for chilling injury and aroma compounds (Table 3). Estragole was of particular interest as it is responsible for licorice aroma in basil [21]. A QTL on LG1 was significant for estragole and significant in

the MQM of two of the chilling assay replicates, however the regions did not overlap and no relevant candidate genes were identified via annotation. Compounds other than those examined in this study may still be associated with chilling response in basil, such as eugenol [92, 93]. Eugenol levels in this population were too low to assess for QTL and correlations with chilling response, but it could play a role in other populations of basil with higher endogenous concentrations.

This paper presents the potential for development of chilling tolerant sweet basils, which do not yet exist in the market. A cold-tolerant evergreen hybrid produced from the interspecific cross of *O. kilimandscharicum* and *O. basilicum* was reported as being cold tolerant [94]. Although this study did not report the aroma or appearance of the hybrid, it is likely high in camphor and ornamental in appearance like the *O. kilimandscharicum* parent. This is the case with 'African blue' basil, which is an interspecific cross from the same two species (*O. kilimandscharicum* x *O. basilicum* 'Dark Opal'). Furthermore, this variety is sterile, which inhibits commercial seed production and breeding for improved aroma [95] unless embryo rescue or other techniques are used to restore fertility. 'Magic Mountain' (Ball Seed) claims to be less cold-sensitive than other basils. However, the aroma and appearance of 'Magic Mountain' is not distinctly Genovese-style and 'CB15' was significantly more cold tolerant in our evaluations. A newer ornamental basil called 'Christmas basil' (Thompson & Morgan; Luv2Garden) has been reported as "half-hardy", yet this basil is ornamental in appearance with aroma consists of pine, fruit or mulled wine instead of sweet basil. Future plans include stacking traits of interest with knowledge of the QTLs to develop chilling tolerant sweet basil varieties.

## Supporting information

**S1 Table. Annotations within each QTL associated with cold response or aroma profile.**
The QTLs are represented by their linkage group (LG) number.
(XLSX)

**S1 Fig.** Flow cytometry for the parents (A-B), hybrid (C), and selected F2 progeny (D-F). Estimated genome sizes of the accessions were 4.85–4.88 Gb (2C DNA content) with support for tetraploid inheritance.
(PDF)

**S2 Fig. Heatmap of the marker-pairwise estimated recombination fractions vs.** LOD scores.
(PDF)

## Acknowledgments

Special thanks to the Rutgers New Use Agriculture and Natural Plant Products Program, to the original work of Paulo Ribeiro and Robert Pyne while working on their dissertations on basil in the Simon laboratory at Rutgers University. We wish to thank Jennifer Vaiciunas and Christine Kubik for their instruction and assistance in the preparation of ddRADseq libraries. We thank Alexander Barrett, Trevor Styles, Utsav Kumar, Elizabeth Kogan, Elizabeth Peters, Shakthi Rave, Sophia Hsueh, Asma Khan and Jeremy Simon for their assistance on data collection, Dr. Itay Gonda and Dr. Rong Di for their fruitful discussions, and our greenhouse staff, including Adam Morgan, Joe Florentine, Jeffrey Akers and Amy Abate for their plant maintenance support.

## Author Contributions

**Conceptualization:** Lara J. Brindisi, James E. Simon.

**Data curation:** Lara J. Brindisi, Robert Mattera, III, Sonika Mudiyala.

**Formal analysis:** Lara J. Brindisi.

**Funding acquisition:** Lara J. Brindisi, James E. Simon.

**Investigation:** Lara J. Brindisi, Joshua Honig, James E. Simon.

**Methodology:** Lara J. Brindisi, Joshua Honig, James E. Simon.

**Project administration:** Lara J. Brindisi, James E. Simon.

**Resources:** James E. Simon.

**Software:** Lara J. Brindisi, Robert Mattera, III.

**Supervision:** Lara J. Brindisi, James E. Simon.

**Validation:** James E. Simon.

**Visualization:** Lara J. Brindisi.

**Writing – original draft:** Lara J. Brindisi.

**Writing – review & editing:** Lara J. Brindisi, Robert Mattera, III, Joshua Honig, James E. Simon.

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
