## [Decision Letter · Decision Letter 0]

19 Dec 2023

PONE-D-23-32565Genetic linkage mapping and quantitative trait locus (QTL) analysis of sweet basil (Ocimum basilicum L.) to identify genomic regions associated with cold tolerance and major volatilesPLOS ONE

Dear Dr. Simon,

Thank you for submitting your manuscript to PLOS ONE. After careful consideration, we feel that it has merit but does not fully meet PLOS ONE’s publication criteria as it currently stands. Therefore, we invite you to submit a revised version of the manuscript that addresses the points raised during the review process.

We look forward to receiving your revised manuscript.

Kind regards,

Ahmed E. Abdel Moneim

Academic Editor

PLOS ONE

“This research was supported by Research Grant No. IS-5292-20 RF from BARD, the United States - Israel Binational Agricultural Research and Development Fund (www.bard-isus.com/), and the New Jersey Agricultural Experiment Station and HATCH project 12170 (www.nifa.usda.gov/grants/programs/hatch-act-1887-multistate-research-fund) awarded to James E. Simon (JES).”

4. PLOS requires an ORCID iD for the corresponding author in Editorial Manager on papers submitted after December 6th, 2016. Please ensure that you have an ORCID iD and that it is validated in Editorial Manager. To do this, go to ‘Update my Information’ (in the upper left-hand corner of the main menu), and click on the Fetch/Validate link next to the ORCID field. This will take you to the ORCID site and allow you to create a new iD or authenticate a pre-existing iD in Editorial Manager. Please see the following video for instructions on linking an ORCID iD to your Editorial Manager account: https://www.youtube.com/watch?v=_xcclfuvtxQ.

Reviewers' comments:

Reviewer's Responses to Questions

**Comments to the Author**

1. Is the manuscript technically sound, and do the data support the conclusions?

Reviewer #1: Yes

Reviewer #2: Yes

2. Has the statistical analysis been performed appropriately and rigorously? 

Reviewer #1: No

Reviewer #2: Yes

3. Have the authors made all data underlying the findings in their manuscript fully available?

Reviewer #1: Yes

Reviewer #2: No

4. Is the manuscript presented in an intelligible fashion and written in standard English?

Reviewer #1: Yes

Reviewer #2: Yes

5. Review Comments to the Author

Reviewer #1: Brindisi et al. conducted genetic mapping and identified the QTLs associated with cold tolerance in Sweet Basil and major volatiles. The methodologies are are correctly applied and the manuscript is well written. I have few suggestions or queries to improve the manuscript further,

1. Please give tolerance levels of both parents used in developing the RIL population.

2. In abstract-I dont see what is the QTL Ch-24 means? What to predict from sentence: Ch-24 represented the QTL with the largest effect and was significant in all three replicates. As I can see in results this QTL is identified for chilling.

3. Introduction looks significantly longer

4. Table 1. Please check literature for QTL nomenclature. The QTL naming pattern is doubtful. the QTL names starts with 'q'.

5. Table 2. It is good that you have provided the table for the genes and annotations. However, it hard to understand in which QTL regions they are identified. You can include a column for QTL region.

6. In Figure 8. The marker is not highly polimorphic.

7. Overall figures should be revised. Please include a normal distribution plot.

Reviewer #2: The manuscript “Genetic linkage mapping and quantitative trait locus (QTL) analysis of sweet basil (Ocimum basilicum L.) to identify genomic regions associated with cold tolerance and major volatiles” is an good addition in a genetics of basil to understand cold tolerance mechanism and volatiles composition. The authors performed robust experiments and produced reliable data. The manuscript could be accepted in PLOS ONE after addressing minor comments given below.

Line 42-44: What does it mean when the authors used term “Significant QTLs”?

Line 40-41: We constructed/developed genetic linkage map and performed QTL analysis to identify genomic regions

Line 41: “2” should be subscript

Line 123: Either use both pathogens names and diseases name. Correct it throughout the manuscript

Line 136-137: Mention in bracket that you are going to use short form of the line name

Line 173-177: Mention the instruments makers and where those facilities were used.

Line 220: Remove the Genotyping-by-sequencing (GBS) word from the start of the sentence. It is confusing GBS with RADseq. Those are two similar yet different methods

Line 288-298: Just refer the table for the key words

Line 358-359: There should be some threshold for the dissertation percentage? Authors could have removed the extremely distorted markers from the map.

Line 350-360: The Seq data should be publicly available.

Line 361-362: Where is the heatmap?

6. PLOS authors have the option to publish the peer review history of their article (what does this mean?). If published, this will include your full peer review and any attached files.

Reviewer #1: **Yes: **Sunil S. Gangurde

Reviewer #2: **Yes: **Muhammad Irfan Siddique

---

## [Author Response · Author response to Decision Letter 0]

10 Jan 2024

Dear Editor,

We thank you and the reviewers for the careful comments, critiques and questions. We have revised our originally submitted manuscript addressing each point as discussed below. These changes have strengthened significantly and improved the manuscript. Thank you. We are most appreciative and now hope our detailed responses below coupled with the revisions more than meet your expectations to accept and meet the publication guidelines. Do let us know if you have further questions. 

Thank you.

Jim Simon and Co-authors.

January 9, 2023

Responses to Reviewer #1:

1. Please give tolerance levels of both parents used in developing the RIL population.

Our response: Tolerance and resistance levels of the inbred parents and F2 progeny are described in terms of % necrosis in lines 298-395 and illustrated in Figures 1-2. The method of collection is described in lines 159-174. 

2. In abstract-I dont see what is the QTL Ch-24 means? What to predict from sentence: Ch-24 represented the QTL with the largest effect and was significant in all three replicates. As I can see in results this QTL is identified for chilling.

Our response: Thank you for pointing this out. This has been corrected.

3. Introduction looks significantly longer

Our response: We find that the discussion in the introduction is necessary to provide background for the study. We welcome any suggestions for specific sections that should be re-examined but ask that it be kept as is.

4. Table 1. Please check literature for QTL nomenclature. The QTL naming pattern is doubtful. the QTL names starts with 'q'.

Our response: We agree and replaced all names to follow a more suitable naming convention of qCH#.

5. Table 2. It is good that you have provided the table for the genes and annotations. However, it hard to understand in which QTL regions they are identified. You can include a column for QTL region.

Our response: There is one QTL per linkage group and we have included the LG in the leftmost column. Upon further inspection, we believe this is clear to our readers.

6. In Figure 8. The marker is not highly polimorphic.

Our response: Yes, we agree. This could be due to a few reasons. These markers are nearest to the peak detected by the QTL but may not be the responsible marker for the phenotype, especially with the size of the peak and marker density. Further, the trait is a highly complex trait and we expect more variation in the population than a qualitative or Mendelian trait. This is discussed in the paper.

Responses to Reviewer #2:

1. Line 42-44: What does it mean when the authors used term “Significant QTLs”

Our response: We addressed this by adding a definition “QTLs were deemed statistically significant when the LOD exceeded the significance threshold” to lines 325-326. “Statistically” significant was also added to this line and other areas for clarity.

2. Line 40-41: We constructed/developed genetic linkage map and performed QTL analysis v. Line 41: “2” should be subscript

Our response: Thank you. This has been corrected.

Line 123: Either use both pathogens names and diseases name. Correct it throughout the manuscript

Our response: Revised to list full pathogen names for “DMR” and “FOB” once throughout manuscript.

3. Line 136-137: Mention in bracket that you are going to use short form of the line name

Our response: Corrected.

4. Line 173-177: Mention the instruments makers and where those facilities were used.

Our response: Details added.

5. Line 220: Remove the Genotyping-by-sequencing (GBS) word from the start of the sentence. It is confusing GBS with RADseq. Those are two similar yet different methods

Our response: Corrected.

6. Line 288-298: Just refer the table for the key words

Our response: Corrected. Lines removed from manuscript and added to a sheet on the supplementary table. 

7. Line 358-359: There should be some threshold for the dissertation percentage? Authors could have removed the extremely distorted markers from the map.

Our response: Yes, we acknowledge that removal of distorted markers would normally occur in linkage mapping. However, in this case, the extremely distorted markers could not be removed as a majority of the makers were distorted despite the threshold (see line 368). We find this phenomenon to be most interesting and thus extensively discuss possible reasons for this occurrence in the discussion (see lines 496-520). Instead of setting an arbitrary threshold and removing most of the markers, we took the opportunity to explore how these results could be linked to allopolyploidy, which highlights the novelty of this research and focuses on the unique genetic issues one faces with such plant species. We are continuing to explore how to dissect the reference map to further explain the marker segregation, but this is currently outside the scope of this research paper. We look forward to elucidating this further in the future. 

8. Line 350-360: The Seq data should be publicly available. 

Our response: Thank you for addressing this. We will add this data as a supplementary document.

9. Line 361-362: Where is the heatmap?

Our response: Thank you for addressing this. We have added this data as a supplementary document.

---

## [Decision Letter · Decision Letter 1]

16 Feb 2024

Genetic linkage mapping and quantitative trait locus (QTL) analysis of sweet basil (Ocimum basilicum L.) to identify genomic regions associated with cold tolerance and major volatiles

PONE-D-23-32565R1

Dear Dr. Simon,

We’re pleased to inform you that your manuscript has been judged scientifically suitable for publication and will be formally accepted for publication once it meets all outstanding technical requirements.

Kind regards,

Ahmed E. Abdel Moneim

Academic Editor

PLOS ONE

Additional Editor Comments (optional):

Reviewers' comments:

Reviewer's Responses to Questions

**Comments to the Author**

1. If the authors have adequately addressed your comments raised in a previous round of review and you feel that this manuscript is now acceptable for publication, you may indicate that here to bypass the “Comments to the Author” section, enter your conflict of interest statement in the “Confidential to Editor” section, and submit your "Accept" recommendation.

Reviewer #1: All comments have been addressed

2. Is the manuscript technically sound, and do the data support the conclusions?

Reviewer #1: Yes

3. Has the statistical analysis been performed appropriately and rigorously? 

Reviewer #1: Yes

4. Have the authors made all data underlying the findings in their manuscript fully available?

Reviewer #1: Yes

5. Is the manuscript presented in an intelligible fashion and written in standard English?

Reviewer #1: Yes

6. Review Comments to the Author

Reviewer #1: (No Response)

7. PLOS authors have the option to publish the peer review history of their article (what does this mean?). If published, this will include your full peer review and any attached files.

Reviewer #1: **Yes: **Sunil S Gangurde

---

## [Editor Report · Acceptance letter]

25 Mar 2024

PONE-D-23-32565R1 

PLOS ONE

Dear Dr. Simon, 

I'm pleased to inform you that your manuscript has been deemed suitable for publication in PLOS ONE. Congratulations! Your manuscript is now being handed over to our production team.

Kind regards, 

on behalf of

Dr. Ahmed E. Abdel Moneim 

Academic Editor

PLOS ONE